# Group Changes in Cortisol and Heart Rate Variability of Children with Down Syndrome and Children with Autism Spectrum Disorder during Dog-Assisted Therapy

**DOI:** 10.3390/children10071200

**Published:** 2023-07-11

**Authors:** Richard E. Griffioen, Geert J. M. van Boxtel, Theo Verheggen, Marie-Jose Enders-Slegers, Steffie Van Der Steen

**Affiliations:** 1Department of Animal Assisted Interventions, Aeres University of Applied Sciences, De Drieslag 4, 8251 JZ Dronten, The Netherlands; r.griffioen@aeres.nl; 2Tilburg School of Social and Behavioral Sciences, Tilburg University, Professor Cobbenhagenlaan, 5037 AB Tilburg, The Netherlands; g.j.m.vboxtel@tilburguniversity.edu; 3Department of Psychology and Education, Open University The Netherlands, Valkenburgerweg 177, 6419 AT Heerlen, The Netherlands; theoverheggen@gmail.com; 4Faculty of Behavioural and Social Sciences, University of Groningen, Grote Kruisstraat 2/1, 9712 TS Groningen, The Netherlands; s.van.der.steen@rug.nl

**Keywords:** animal-assisted interventions, HRV, cortisol, stress, dog-assisted therapy, Down syndrome, autism spectrum disorder

## Abstract

Dog-assisted therapy is hypothesized to lower stress in children with autism spectrum disorder (ASD) and children with Down syndrome (DS), which may be visible on a physiological level. In this study, we measured heart rate variability (HRV) and salivary cortisol of 20 children with DS or ASD at the beginning and end of six weekly sessions of dog-assisted therapy. We found a decrease of cortisol levels during single sessions, but no overall effect after six sessions (six weeks). The effect of dog-assisted therapy on the increase of HRV could not be confirmed. This study is one of the first to use physiological measurements to test the effects of DAT.

## 1. Introduction

Children with Down syndrome (DS) and children with autism spectrum disorder (ASD) often experience stress due to difficulties with their social interactions and verbalization, e.g., when initiating and maintaining social relationships or when they have difficulties with cognitive switching [1,2,3]. These children seem more fearful than their typically developing peers [4], use less effective coping strategies in response to stressful situations [5], and have strong stress reactions to their social environment. Children with ASD or DS show an increased risk of experiencing traumatic events, especially social mistreatment [6]. For children and youth with DS, anxiety-related disorders are one of the most common co-occurring diagnoses [7].

Some lines of evidence suggest that interactions with animals, specifically dogs, may be stress-reducing for several populations [8,9]. First, various studies have revealed the positive effects of dog ownership on family functioning, as well as decreased child anxiety and stress [10,11,12]. Typically developing children who are supported in a stressful situation by their own dog experience less stress (as measured by their self-reported feelings of stress and salivary cortisol) than children who are alone in the same situation, or who are supported by their parents [13]. Dogs can also offer support in case of developmental or emotional problems [14,15]. Beetz and colleagues [16,17] measured the cortisol levels of children with insecure attachment styles in the presence of dogs. The support of a dog during the study was associated with significantly lower cortisol levels. This effect was strongly correlated with the time the children spent in physical contact with the dog. There is also some evidence that children admitted to a hospital benefit from interactions with dogs, especially with regard to their experienced anxiety and pain [18,19,20]. However, a critical review concluded that most studies on human–animal interactions in hospital settings suffer from weak methodology and flagged construct validity [21]. 

The positive effects of the interactions with dogs on stress symptoms discussed here have mostly been studied in settings in which the children’s families were pet owners. Yet, stress reduction has also been an important hypothesized effect of animal-assisted therapy, and, specifically, dog-assisted therapy (DAT). DAT consists of structured one-on-one sessions, offered by trained professionals who work with certified therapy dogs. Much like other forms of therapy, the child visits the therapist and dog for a number of sessions, in which he/she works on specific therapeutic goals. For children with DS or ASD, DAT has been found to improve their social interactions and communication with other people [22,23] and their abilities to cope with stressful situations [17,24,25]. A recent study on DAT [26] found a positive effect on behavioral synchrony, and a non-significant trend of decreased problem behavior in children with DS and children with ASD. A literature review [24] found two case studies in which stress and arousal were *physiologically* measured in children with ASD during animal-assisted therapy with horses [27] and guinea pigs [25]. A recently published study on the effect of dog-assisted interventions in a classroom context for children with special educational needs found that the dogs seemed to reduce the physiological stress levels of the children [28]. To the best of our knowledge, no studies involving physiological measures on children with DS or ASD in a dog-assisted therapy context have been conducted.

### Physiological Measurements of Stress

A relatively inexpensive and easy way to collect a biomarker of stress is salivary cortisol, which can safely be used in research with children. Research shows a relationship between stress and the hormone cortisol: The more stress a child experiences, the higher its cortisol levels become [29,30,31,32,33]. The hypothalamus–pituitary–adrenocortical axis (HPA) is involved in activating the stress response and plays an important role in physiological coping [33,34]. As indicated by Hanrahan [34], a wide range of factors should be considered while using this measure in research with children, including standardizing the time for sample collection, including baseline samples, using consistent collection materials and methods, controlling for certain drinks, foods, medications, and diagnoses, and establishing sound procedures and protocols.

In addition to cortisol, heart rate variability (HRV) is also a viable physiological indicator of stress [29,35,36]. HRV refers to the variation in intervals between successive heartbeats (IBIs: inter-beat intervals) and is an indicator of the functioning of the autonomic nervous system. The autonomic nervous system (parasympathetic and sympathetic branches) is the part of the nervous system that controls the visceral functions of the body, including the heart, the movements of the gastrointestinal areas, and many other vital activities. Mental and emotional feelings directly influence the autonomic nervous system. Research shows that high HRV is related to rest and relaxation, while low HRV is related to effort and stress [35,36]. HRV measurements identify the activity of the parasympathetic system by measuring the beat-to-beat alteration in heart rate; e.g., low HRV is indicative of reduced parasympathetic cardiac control [37,38].

In summary, children with ASD or DS experience stress in various situations. Dog-assisted therapy (DAT) may offer relief, which might be visible on a physiological level, but little research has been carried out in a therapy context with dogs and within these two client populations. This research therefore set out to investigate the following two research questions: (1) To what extent can we observe differences in the HRV measurements and cortisol levels of children with DS or ASD after following a 6-week DAT program? and (2) To what extent do these HRV measurements and cortisol levels change *during* DAT sessions (comparing the measurements before and after the session)?

## 2. Materials and Methods

### 2.1. Ethical Considerations

In this explorative study, we observe how dog-assisted therapy (DAT) influences the experienced stress of children with DS and ASD. Given that this is an explorative study—the first in which physiological measures are used in a DAT context with these client populations—and because we work with a vulnerable group of children, we have chosen not to use a (non-intervention) control group at this point.

In this study, twenty children with DS or ASD were followed for six weeks while they received dog-assisted therapy (DAT). At the start of the study, written consent was obtained from the parents of the children. The parents were also informed about the aim of the study, as well as about the protocol of the therapy. It was made clear to them that they were free to stop their participation at any moment. During the research, the dog’s welfare was continually monitored by a handler. All necessary precautions were taken to prevent possible harm to the dogs. The Medical Ethical Review Committee of the University of Amsterdam stated that an official approval of this study was not required concerning the Medical Research Involving Human Subject Act (WMO).

### 2.2. Design

The current study investigates the effect of a 6-week DAT program on the salivary cortisol levels and heart rate variability of 20 children with DS or ASD who received a weekly one-hour session of dog-assisted therapy. Before and after every session we collected salivary cortisol and recorded children’s heart rate variability.

### 2.3. Participants

Twenty children volunteered in this study, eight children with Down syndrome (DS; six males and two females with a mean age of 12, range 10–14 years) and twelve children with autism spectrum disorder (ASD; eight males and four females with a mean age of 13, range 11–15 years). All children were checked by a pediatrician prior to the research, and had no psychiatric co-morbidity or suffered from heart dysfunctions. All children who participated in the study had a total IQ score above 40 (which makes them eligible for enrollment in special education in the Netherlands) and were able to make themselves understood using at least single words or short sentences. All children received their education at a school for students with special needs. No allergies and medication use were reported by their parents.

The children were recruited through an invitation letter to the parents in the network of the first author, through the website of the SAM Foundation (www.stichtingsam.nl (accessed on 15 May 2023)), through an organization for therapy dogs (www.contacthond.nl (accessed on 15 May 2023)), and by referrals from healthcare professionals. All parents signed an informed consent document and agreed with the procedure. Exclusion criteria were fear of dogs, severe dog allergies, wheelchair dependency, and reported previous aggression toward animals.

### 2.4. Procedure

The sessions with the dog took place at a gymnasium. The children were accompanied by a therapist. They worked together in a 30 min session, once a week for a period of 6 weeks. Activities during the sessions were selected from the CTAC method [39]. We selected psycho-motor and socialization activities in particular, for example, having the dog follow the child’s movements and letting the child exercise his/her balance and be aware of the effect of posture and expression on the dog. The therapist explained what was expected of the child and which gestures and encouraging words were needed to work with the dog. All therapy dogs within this study came from organization Contacthond and were extensively tested and trained before they became part of the therapy setting. Among the requirements were a good physical overall health, being vaccinated, and being at least 2 years old. In addition, all dogs went through a series of behavioral tests conducted by experienced trainers, testing whether they react calmly to new situations and assessing their temperament. The therapy dogs were always supervised within the therapy setting by an experienced and certified dog handler employed by the Contacthond Foundation. Based on character and temperament, we connected a specific therapy dog to each client for the full duration of the therapy.

During the first phase of each session, the child and the dog, guided by the therapist and dog handler, participated in some exercises to get used to performing the tasks. The second phase of each session was characterized by encouraging the child to build an obstacle course, and to lead the dog through a series of obstacles asking the dog to perform certain commands, such as sitting and lying on a mat, slalom-walking around cones and walking on a bench, or jumping over a low bar [26].

### 2.5. HRV Measurements

At the beginning of each session, the children were asked to put on a chest belt with heart rate sensor and wear a Polar device V800 wristwatch (Polar Electro Oy, Kempele Finland) to record N–N intervals (inter-beat intervals). The data were read into Polar FlowSync software after the sessions. The HRV of the children (*n* = 20) were registered during two periods of 3 min, before and at the end of each session. During this registration, the children were asked to sit on a chair. Based on visual inspection, we chose the two artifact-free periods. The artifacts were due to missing beats by the chest belt and sudden movements of the children.

**Analysis method HRV—Time domain:** We first analyzed the global indicators of HRV. The analyzed time domain HRV parameters are the following: average of the N–N intervals (Mnn) and the standard deviation of the N–N intervals (SDnn) in seconds. The latter is the most straightforward measure of overall HRV. We also calculated the root mean square of successive differences (RMSSD), which is the standardized average difference between consecutive beats. RMSSD is the most commonly used time domain measure for short (beat-to-beat) variations.

**Analysis method HRV—Frequency domain:** A more in-depth analysis of HRV was performed using the frequency bands. To be specific, we analyzed the low-frequency (LF) band: 0.04–0.15 Hz (period of 6.7–25 s). This “blood pressure band” is associated with a relatively large contribution from the sympathetic branch of the autonomic nervous system. The high-frequency (HF; 0.15–0.4 Hz; period of 2.5–6.7 s) “respiratory band”, associated with a relatively large contribution from the parasympathetic branch of the autonomic nervous system, was also analyzed. Lastly, we used the LF/HF ratio, which is often used as index for the relative activation of the sympathetic versus the parasympathetic nervous system [37].

**HRV data preparation.** The HRV data were processed in 3 steps: pre-processing, data analysis, and statistical analysis. The first step, pre-processing, was intended to reach correct files with N–N intervals (inter-beat intervals, IBIs) for each child, for each session and for the intervals within a session. Some children had fewer sessions because of insufficient artifact-free data, or had missed sessions due to illness. Secondly, the actual data analysis was performed. For the raw N–N intervals, artifact correction was applied. When a particular IBI was outside the range of + or −3 standard deviations (SD) around the average, it was removed from the series and interpolated by means of a cubic spline interpolation. In total, there were 144 measurements and the intervals averaged 209.1 ms (min. 34.6–max. 319.5 s, SD = 76.5 ms).

**HRV statistical analysis:** The root mean square of successive differences (RMSSD), the mean R–R (Mnn), and standard deviation of N–N intervals (SDnn) were computed in the time domain, and low-frequency band (LF, 0.04–0.15 Hz), high-frequency band (HF, 0.15–0.4 Hz), and the LF/HF ratio were computed from the Lomb–Scargle spectra. The data of SDnn were not normally distributed. We solved this by applying a logarithmic transformation. The log-transformed data were then analyzed. All variables obtained were analyzed using mixed-effect (non-linear mixed-effects) models. This was carried out with the R-statistical package version 3.5.2. NLME models are able to handle missing data, and are able to decrease the residual variance in comparison with a regular repeated-measures ANOVA.

### 2.6. Cortisol Measurements

Samples of salivary cortisol were collected from the children prior to and at 10 min after sessions were completed, at a similar time of day (9.45 a.m., start session 10 a.m.) for each participant during six weeks. The collection was standardized using a Salivette (Sarstedt AG & Co, Numbrecht, Germany), a cotton roll packaged in a plastic tube-like container, and the procedure was carried out by the same assistant every time. The child had to chew on the cotton roll for 2 min. We controlled for foods and drinks for 1 h before the samples (no caffeinated products, no dairy products, and no steroids). In addition, children did not eat or drink for half an hour before each session. The samples were refrigerated within 1 h after collection. After collecting samples for six weeks, we sent the samples to the laboratory of the Wilhelmina University Children Hospital of Utrecht, the Netherlands for analysis. Cortisol in saliva was measured without extraction using an in-house competitive radio-immunoassay employing a polyclonal anti-cortisol antibody (K7348). [1,2-^3^H(N)]-Hydrocortisone (PerkinElmer Inc., Groningen, The Netherlands, NET396250UC) was used as a tracer. The cortisol samples obtained were also analyzed using mixed-effect (LME) models with the R-statistical package version 3.5.2.

## 3. Results

### 3.1. Heart Rate Variability, Time Domain

The overall average length of the N–N intervals (Mnn) was 0.69 s (SE 0.11 s). This corresponds to an average heart rate of 86.30 bpm. The averages were fairly normally distributed. Mnn does seem to decrease slightly in the third session, but a visual inspection of Figure 1 shows no clear trend in Mnn over the course of six sessions. Although there seemed to be no difference over all sessions, the Mnn before and after a session did vary. Figure 1 shows that the Mnn are somewhat higher at the beginning of each session, meaning that the heart rate is slightly lower at the beginning.

The standard deviations of the R–R intervals (SDnn) provide a global measure for HRV in the time domain. The average SDnn was 0.078 (SE 0.01). Again, no clear overall trend could be observed over the six sessions, although a visual inspection of Figure 2 shows that the SDnn seemed to decrease from the first to third session and then increased to a somewhat stable level over the last three sessions. The SDnn at the end of each session seemed to be lower than at the beginning of each session, indicating less variability after each single session. The root mean square of successive differences (RMSSD) shows roughly the same picture, namely, no clear trend over sessions, and a decrease of the heart rate variability within sessions (see Figure 3).

Table 1 shows the NLME model results for the time domain measures (Mnn, LogSDnn, and RMSSD). No significant changes in the time domain measures were found over the six sessions. However, with regard to measures taken within a session, we found significant changes in Mnn (*F*(1, 113) = 12.86, *p* < 0.01), SDnn (*F*(1, 113) = 5.45, *p* = 0.02) and RMSSD (*F*(1, 113) = 12.96, *p* < 0.01). No interaction effects were found.

### 3.2. Frequency Domain

With regard to the low-frequency band (LF), no trend could be observed over the course of six sessions, as well as no clear trend within sessions. The high-frequency band (HF) showed a similar pattern as the time domain measures, and especially the RMSSD pattern. Although no clear trend over the six sessions could be observed, HF seemed to decrease during Sessions 1, 5, and 6, and showed a very subtle (likely negligible) decrease in Session 3 (see Figure 4).

Table 2 shows the NLME model results for the frequency domain measures (LF band, HF band, and LF/HF ratio). No significant findings for the LF band were observed. The within-session trend that could be observed for the HF band fell short of significance (*F*(1, 113) = 3.38, *p* = 0.07). The within-session LF/HF ratio did show a significant change (*F*(1, 113) = 4.96, *p* = 0.03). The interaction effect (session * within session of LF/HF) also showed a significant change (*F*(5, 113) = 3.07, *p* = 0.01) which indicates that the within-session effects of the LF/HF ratio differed across the sessions. This is also visible in Figure 5. The changes in the LF/HF ratio within the fifth and sixth session were bigger and in a different direction compared to the LF/HF ratio changes within the other sessions.

### 3.3. Cortisol Measures

We found no clear trend with regard to the changes in cortisol levels over the six sessions (see Figure 6). Cortisol at the beginning of a session alternately decreased and increased over the course of six weeks, and cortisol levels at the end of a session decreased from the first to fourth session, and then increased again in the last two sessions. Yet, the cortisol levels measured within a session were always lower at the end of each session compared to the levels at the start of the session. In line with the visual inspection of Figure 6, Table 3 shows no significant effect of cortisol across sessions, but a significant within-session effect (*F*(1, 159) = 25.36, *p* < 0.01). No significant interaction effect was found.

## 4. Discussion

In this study, we analyzed physiological changes in 8 children with Down syndrome (DS) and 12 children with Autism Spectrum Disorder (ASD) who took part in a 6-week dog-assisted therapy (DAT) program. The children worked with the same dog during the 6-week program. To be specific, we measured heart rate variability (HRV) and salivary cortisol at the beginning and end of six weekly sessions of DAT. To our knowledge, only one study measuring cortisol during a dog-assisted intervention in a special education classroom setting was recently published [28], which makes this the first study in which two physiological measures (HRV and cortisol) are collected during a DAT program for children with ASD and DS. Yet, note that in two previous (case) studies on animal-assisted therapy with horses [27] and guinea pigs [25], physiological measures were also collected.

Our results show that both HRV (time and frequency domain) and cortisol measures decreased significantly within each session. Over the course of the full program (six sessions), however, no significant changes in HRV and salivary cortisol could be found. In addition, the data mostly showed no significant interaction effects, apart from the LF/HF ratio, which showed a different within-session effect across sessions. That is, the changes in the LF/HF ratio within the fifth and sixth session were bigger compared to the changes within the first four sessions.

Interestingly, there seems to be a discrepancy between the two types of physiological data. To be specific, the time-domain HRV measurements decrease within sessions, which generally indicates a higher level of arousal. At the same time, the cortisol levels decrease within sessions, which corresponds to *lower* levels of arousal. A reason for this discrepancy might be that, during the course of the session, the movement activity of the children increased, which makes the heart rate increase as well. Although the children were seated on a chair when taking the HRV measurements, it takes time for an increased heart rate to return to a baseline level. It is well-known that HRV decreases when the heart rate is increased [40].

This means that the higher activity levels within DAT sessions might have confounded the HRV results. From the HF frequency results, it also appears that the children were breathing faster over the course of a session. Yet, since breathing is not directly measured, and since this interpretation of the HF variance is not undisputed [37], we cannot state this with confidence. Cortisol levels do not depend on movement, provided that the movement is not caused by a flight reaction [41]. Cortisol levels are, therefore, possibly a more reliable measure of stress reduction during an active form of therapy such as DAT.

To our knowledge, HRV has not previously been measured in similar contexts. Although research on cortisol changes during DAT is scarce, it has been measured in pet owners [42], as well as in a study in which children with insecure attachment styles responded to a stressful situation in the presence of a dog [16]. The results of those studies are in line with this study, i.e., less arousal in the presence of a dog. Yet, a study on salivary cortisol nerve growth factor and salivary alpha amylase (also two indicators of perceived stress) levels taken from students during their final exams in the presence of a dog showed no detectable differences, even though students’ own perceptions of their stress levels decreased [43]. This discrepancy between studies shows that we should be cautious when interpreting cortisol levels in the context of animal-assisted interventions. Based on the current study and previous studies [16,42,43], we can cautiously conclude that cortisol levels decrease on the short term (during a session) when interacting with animals.

Contrary to our expectations, there were no significant HRV and cortisol changes over the full course of a 6-week DAT program. Although this study lacked a control group, we compared the physiological measures of the children to their levels before the beginning of the therapy, with the very first measurement taken before the start of the first session. Although we cannot draw firm conclusions, we do know that, for this group of children, this DAT program did not succeed in bringing children to lower stress levels in the longer term (6 weeks). Note, however, that a DAT program of a longer duration or of a higher intensity (several times a week)—thereby increasing the number of child–dog interactions—could possibly lead to different results. This would correspond to the finding that dog owners who have frequent interactions with their dogs appear to have lower cortisol levels [42].

This study has several limitations. First, we decided to include no control group for ethical reasons (i.e., we worked with a vulnerable group of children and there was no precedent). Without doubt, this limits the generalizability of our results. Being an exploratory first study on physiological changes during DAT, this study also has a fairly limited scope, simply assessing whether DAT might influence physiological measures in children with ASD or DS. No extra measures of children’s behavior, for example parental questionnaires or observations, were included. Second, because we worked with a specific group of children, we also needed professionally trained therapists to guide these children. This resulted in a relatively small sample size. Therefore, we recommend conducting this study in a larger sample, possibly combined with a DAT program of a longer duration (see above), as there are some indications from previous studies that prolonged exposure to dogs could influence cortisol levels [42]. Lastly, even though we used an HRV chest strap customized for children, the artifacts in our heart rate recordings could indicate that the fit was suboptimal. As mentioned in the method section, we tried to (partially) solve this by measuring HRV while children sat in a chair before and at the end of each session. Future studies could overcome this by using wrist devices that register HRV, or other equipment that registers respiratory frequencies and blood pressure in addition to HRV. Despite these limitations, we believe that this study sets an important precedent for the use of physiological data, specifically cortisol measures, in research on animal-assisted interventions.

## Figures and Tables

**Figure 1 children-10-01200-f001:**
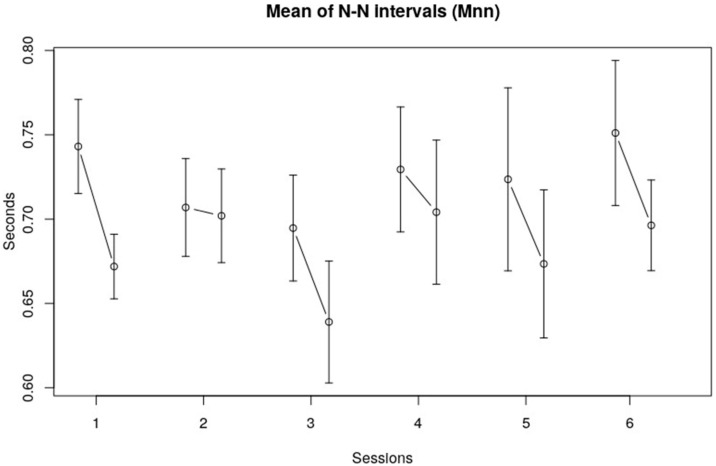
Mean of N–N intervals (Mnn). The vertical axis displays the mean Mnn of the children in seconds; the horizontal axis represents the session number. The error bars in the figure represent the standard error of the mean.

**Figure 2 children-10-01200-f002:**
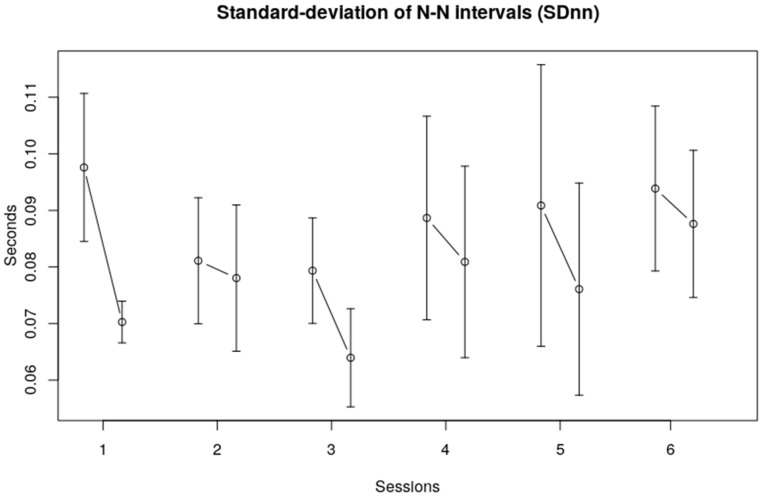
Standard deviation of R–R intervals (SDnn). The vertical axis displays the Sdnn of the children in seconds; the horizontal axis represents the session number. The error bars in the figure represent the standard error of the mean.

**Figure 3 children-10-01200-f003:**
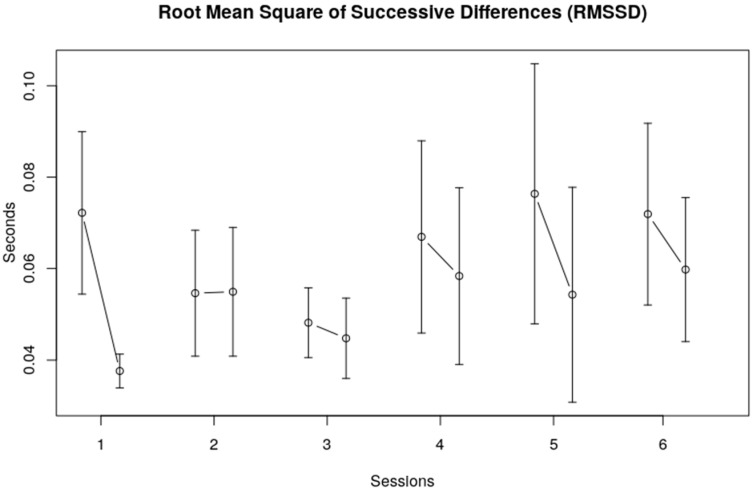
Root mean square of successive differences (RMSSD). The vertical axis displays RMSSD in seconds; the horizontal axis represents the session number. The error bars in the figure represent the standard error of the mean.

**Figure 4 children-10-01200-f004:**
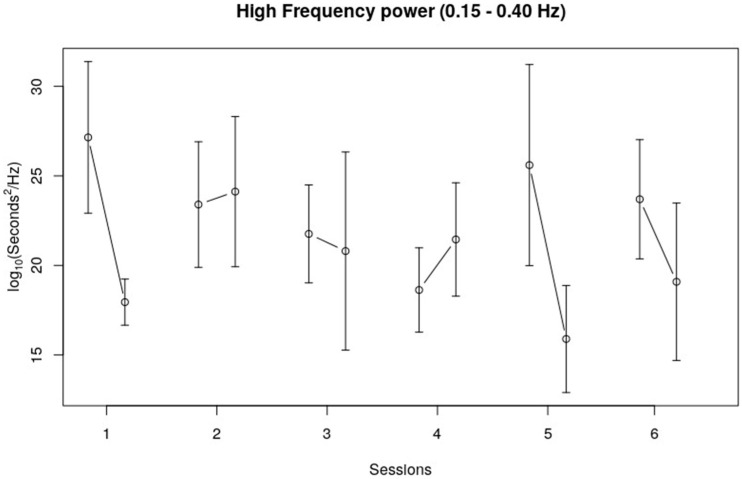
HF band data. (0.15–0.40) The vertical axis displays the HF log-scale in seconds; the horizontal axis represents the session number. The error bars in the figure represent the standard error of the mean.

**Figure 5 children-10-01200-f005:**
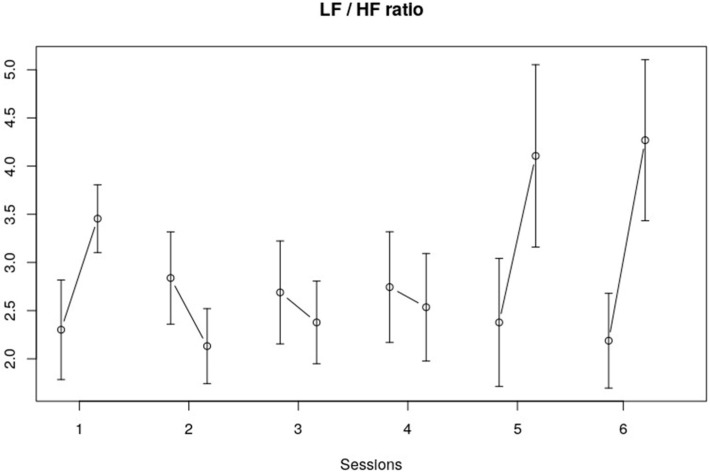
LF/HF ratio. The vertical axis represents the LF/HF ratio in seconds; the horizontal axis displays the session number. The error bars in the figure represent the standard error of the mean.

**Figure 6 children-10-01200-f006:**
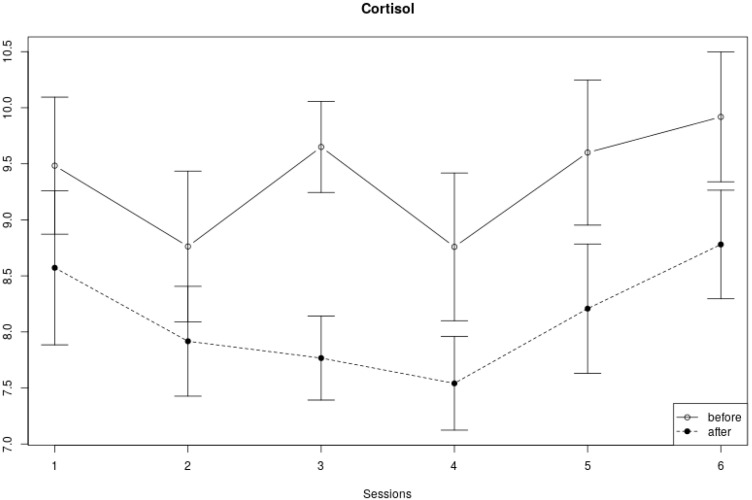
Cortisol measures within and across sessions. The vertical axis displays the cortisol measures in nmol/L, and the horizontal axis represents the session number. The error bars in the figure represent the standard error of the mean.

**Table 1 children-10-01200-t001:** Mnn, LogSDnn, and RMSSD results.

		*F*	df	*p*
Mnn	Session	2.0064	5, 113	0.0830
	Interval within session	12.8576	1, 113	0.0005 *
	Session x within session	0.6853	5, 113	0.6355
SDnn	Session	1.5261	5, 113	0.1873
	Interval within session	5.4518	1, 113	0.0213 *
	Session x within session	0.2777	5, 113	0.9245
RMSSD	Session	1.1591014	5, 113	0.1683
	Interval within session	12.959291	1, 113	0.0005 *
	Session x within session	1.018265	5, 113	0.4104

* *p* < 0.05.

**Table 2 children-10-01200-t002:** NLME model results for the LF band, HF band, and LF/HF ratio.

		*F*	df	*p*
LF	Session	0.32508	5, 113	0.8970
	Interval within session	0.23844	1, 113	0.6263
	Session x within session	1.67228	5, 113	0.1470
HF	Session	0.51381	5, 113	0.7653
	Interval within session	3.37757	1, 113	0.0687
	Session x within session	1.06261	5, 113	0.3850
LF/HF	Session	0.82692	5, 113	0.5331
	Interval within session	4.96285	1, 113	0.0279 *
	Session x within session	3.07045	5, 113	0.0123 *

* *p* < 0.05.

**Table 3 children-10-01200-t003:** NLME model results of the cortisol measures.

		*F*	df	*p*
Cortisol	Over 6 sessions	1.37	5, 159	0.24
	Within-session	25.36	1, 159	<0.0001 *
	Interaction	0.42	5, 159	0.83

* *p* < 0.01.

## Data Availability

Data available on request due to restrictions e.g., privacy or ethical. The data presented in this study are available on request from the corresponding author. The data are not publicly available due to privacy restrictions.

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
