# Peer review of "Group Changes in Cortisol and Heart Rate Variability of Children with Down Syndrome and Children with Autism Spectrum Disorder during Dog-Assisted Therapy"

_children, 2023, doi:10.3390/children10071200_

Round 1
Reviewer 1 Report
This study describes the variation of salivary cortisol levels and HRV in children with Down syndrome and ASD involved in dog-assisted interventions. Please, add "type of the paper".
This is a pilot study without a control group and without baseline data about the two physiological parameters they considered as indicators of stress level. I consider the study design very weak moreover the absence of behavioural analysis associated with physiological parameters is another point that reduces the relevance of the results. Anyway, the authors declare the limitations of the study (this section of the discussion needs to be more exhaustive)
I suggest improving materials and methods with a more detailed description of sessions, settings, activities and dog selection procedures (certification of therapy dogs doesn't exist: certification is a word which has a specific meaning; it's a third-party assessment based on a standard established by a standardization body and this is not the case (line 58).
Explain better inclusion and exclusion criteria of children and their clinical status (did you verify their heart condition? In Down syndrome, heart diseases or dysfunctions are very common).
Reviewer 2 Report
Review of Griffioen et al.’s Changes in Cortisol and Heart Rate Variability of Children with Down Syndrome and Children with Autism Spectrum Disorder during Dog Assisted Therapy, manuscript id: children-2449573.
Griffioen et al. investigate whether interacting with a therapy dog decreases stress in children with either Down syndrome or autism spectrum disorder. This is investigated with two physiological measures of stress – heart rate variability (HRV) and salivary cortisol – before and after each of six sessions with the therapy dog. Their findings are mixed (HRV indicates an increase in stress pre- to post-session while cortisol indicates the opposite) and hard to interpret across sessions due to the lack of a no-treatment control condition which was not included for ethical reasons. Furthermore, the title of the manuscript might lead some readers to believe that results will be presented separately for children with Down syndrome and for children with autism spectrum disorder. However, only results collapsed across the two groups are presented. For these reasons, I believe that the manuscript will be of limited interest to readers of the journal.
Larger Issues:
What do the error bars in figures 1 through 6 show? Do they show the 95% CI, SD, SE? It is difficult to interpret the graphs without knowing what the error bars show. Lines 285-286: “but HF seemed to decrease during four out of six sessions…” I assume that you refer to sessions 1, 3, 5, and 6 in figure 4. Without knowing exactly what the error bars in figure 4 represent, it is difficult to say whether session 3 seems to decrease or not. The means of the pre- and post-sessions appear very similar, certainly within each other’s error bars suggesting that the decreasing trend might be random variation. It might be better to claim that HF seemed to decrease during three of the six sessions.
Given the title of the manuscript and the two different populations that the authors sampled from, I expected to see the authors present results for the two groups – children with Down syndrome and children with autism spectrum disorder. To only present data collapsed across the two groups was surprising and disappointing. While the sample size for each group is small, given that the authors call the study “explorative” (line 108), it would be worth exploring whether the results apply to one group, the other group, or both groups.
The lack of a no DAT control group (because the study is explorative and for ethical reasons) makes it impossible to interpret the across sessions data. One cannot know whether the ups and downs across the sessions are due to there being no long term effect of DAT for these children or whether factors outside the milieu of the study have changed across sessions. Thus, the statements on lines 365-367 are not supported by the data.
Lines 335-337: Be consistent with the conclusion stated on lines 298 to 300: “the changes in LF/HF ratio within the first, fifth and sixth session were bigger and in a different direction compared to the LF/HF ratio changes within the other sessions.”
Lines 341-353. It seems like the authors are discounting the HRV data. If the HRV data is confounded with heart rate at pre- and post-measurements and can’t be believed, why is so much of the manuscript devoted to the HRV data?
Line 360 – I can’t determine which reference citation [42] refers to, but in some of the literature on therapy dogs and exam stress, salivary cortisol is measured immediately after the interaction with the dog. As Griffioen et al. state, salivary cortisol lags environmental events. Did the study in [42] measure salivary cortisol after an appropriate delay? If not, can that be an alternative explanation for why there was a lack of an effect?
Minor Issues:
Citations use the [#] format, but the references do not; the references are in alphabetical order by the first author’s last name. It is impossible to match citations to references.
On lines 120-122, it isn’t clear whether the ethical waiver applies to the human participants (“Medical Research Involving Human Subject Act”), to the dogs who participated in the research, or both. Please clarify. If the waiver applies only to the human participants, did you get approval or a waiver for using the dogs in the research as well?
Lines 237-238: Line 237 looks like the end of one paragraph and line 238 looks like the start of a new paragraph, but they read like a single paragraph.
Line 249: Figure 2 seems to be covering part of a line. The only text on line 249 is “dren in seconds, the”.
Line 298: “This is also visible in Figure 5, that is…” is a comma splice.
Line 306: Place “Cortisol measures” on a line by itself like the HRV heading on line 222.
Line 325-330: This sentence is somewhat awkward. Consider separating it into two or more sentences.
Line 357: “profesionally” is a typographical error.

The quality of language is fine.
Round 2
Reviewer 2 Report
The authors have adequately addressed most of my concerns. There are a few very minor issues left:
Line 161: Should "hearth dysfunctions" be "heart dysfunctions"?
Line 236: "data was" should be "data were" The figure captions need to state what the error bars represent.
Lines 403-405 Barker et al. did not collect salivary cortisol. They collected salivary nerve growth factor and salivary alpha amylase. Thus, the sentence at lines 403-405 is not factually correct.
Language is fine.
Author Response
Reviewer 2
We thank the reviewer for this remark.
- At line 161 we changed “hearth dysfunctions” into “heart dysfunctions”.
- Line 235: we corrected “data was” into “data were” and added what the error bars represent in figure 1 - 6
“The error bars in the figure represent the standard error of the mean”.
- Lines 403-405. Indeed Barker et al. did not collect salivary cortisol. They collected salivary nerve growth factor and salivary alpha amylase.
Thank you for your comment.
According to this information we corrected the sentence. (lines 413-416)
“Yet, a study on salivary cortisol nerve growth factor and salivary alpha amylase (also two indicators of perceived stress) levels taken from students during their final exams in presence of a dog showed no detectable differences, even though students’ own perceptions of their stress levels decreased [42]”.